# Responses of Soil Respiration and Ecological Environmental Factors to Warming and Thermokarst in River Source Wetlands of the Qinghai Lake Basin

**DOI:** 10.3390/biology13110863

**Published:** 2024-10-24

**Authors:** Yanli Yang, Ni Zhang, Zhiyun Zhou, Lin Li, Kelong Chen, Wei Ji, Xia Zhao

**Affiliations:** 1Qinghai Province Key Laboratory of Physical Geography and Environmental Process, College of Geographical Science, Qinghai Normal University, Xining 810008, China; 18909781391@163.com (Y.Y.); zhangni0224@163.com (N.Z.); 13897423633@163.com (Z.Z.); lilinqnu@163.com (L.L.); zhaoxia@qhnu.edu.cn (X.Z.); 2Key Laboratory of Surface Processes and Ecological Conservation on the Tibetan Plateau, Qinghai Normal University, Xining 810008, China; 3National Positioning Observation and Research Station of Qinghai Lake Wetland Ecosystem in Qinghai, National Forestry and Grassland Administration, Haibei 812300, China; 4Lianyungang Academy of Agricultural Sciences, Lianyungang 222006, China; 5Academy of Plateau Science and Sustainability, Plateau Soil Information Science Research Team, Xining 810008, China

**Keywords:** Qinghai–Tibet Plateau, soil respiration, carbon cycle, thermal subsidence, warming

## Abstract

Global climate warming is accelerating changes in permafrost regions, especially on the Tibetan Plateau, where the active layer of permafrost is deepening and thermal subsidence (thermokarst) phenomena are becoming more frequent. These changes have significant effects on the carbon cycle, which plays a critical role in global climate regulation. In this study, we used an Open-Top Chamber (OTC) device to simulate warming (W) and thermal subsidence (RR) conditions in the river source wetlands of the Qinghai Lake basin. By examining soil temperature, moisture, biomass, microbial diversity, and soil respiration, we aimed to uncover the effects of these changes on soil carbon dynamics. Our findings show that warming increased soil temperature and, combined with thermal subsidence, further amplified these effects, especially during colder seasons. Soil moisture was significantly reduced under thermal subsidence conditions, with the combined treatment (RRW) showing the most significant impact. Microbial analysis showed that warming promoted bacterial diversity but reduced fungal diversity in surface soils. Soil respiration, which includes autotrophic and heterotrophic respiration, followed a unimodal seasonal pattern, with warming reducing autotrophic respiration and thermal subsidence inhibiting heterotrophic respiration. Overall, our results indicate that soil respiration in these wetlands is highly sensitive to temperature changes, particularly under thermal subsidence conditions, with a Q10 value of 7.39. This study provides valuable insights into the carbon cycling responses of the Tibetan Plateau wetlands to climate warming, with implications for regional and global carbon budgets.

## 1. Introduction

Since the Industrial Revolution, the widespread burning of fossil fuels and changes in land use have led to a significant increase in greenhouse gas concentrations in the atmosphere [1]. According to the IPCC’s Sixth Assessment Report, the global average temperatures are expected to rise by 1.5 °C or more over the next 20 years [2]. This increase in global average temperature will intensify soil organic carbon decomposition, reduce soil carbon stocks, and increase carbon emissions, ultimately impacting the global carbon balance [3]. In the Northern Hemisphere, permafrost stores 1014 PgC (in the top 3 m), with the Tibetan Plateau permafrost storing between 14 and 46.18 PgC [4,5,6]. Under future climate warming scenarios, permafrost areas are expected to significantly decrease [7], which typically leads to dramatic changes in soil, vegetation, and hydrological processes. Permafrost degradation will result in the melting of underground ice, triggering a series of thermal melt phenomena [8]. In flat areas, these thermal melt phenomena primarily manifest as thermokarst lakes and thermal subsidence, while in sloped areas, they mainly appear as thermal landslides [9]. These changes could stimulate soil carbon release and transform permafrost regions into significant carbon sources [10,11], potentially leading to a positive feedback loop to climate warming.

Soil respiration refers to the release of CO_2_ through respiration processes in the soil [12]. Each year, the CO_2_ flux released to the atmosphere through soil respiration can reach up to 75 × 10^15^ g C yr^−1^, making it a major source of CO_2_ emissions in terrestrial ecosystems. Soil respiration is crucial for regulating soil carbon stocks and the terrestrial carbon cycle, with even minor fluctuations leading to significant changes in atmospheric CO_2_ concentrations, thereby affecting the global carbon balance. Soil respiration is categorized into autotrophic and heterotrophic respiration based on the different sources of CO_2_. Autotrophic respiration mainly refers to the CO_2_ produced by plant root respiration and rhizosphere respiration, while heterotrophic respiration is the CO_2_ generated from microbial respiration and soil animal respiration [13].Numerous studies have explored the response patterns of ecosystem respiration to various environmental factors, including temperature, nutrients, moisture, and grazing [14,15,16,17]. Zou and Peng’s research found that under a 2 °C increase in soil temperature, heterotrophic respiration and total soil respiration significantly increased, while the impact on autotrophic respiration was relatively minor [18,19]. This is related to the melting of permafrost layers and a significant increase in soil moisture effectiveness due to soil warming. A meta-analysis by Lu et al. on ecosystem responses to warming reported that the increase in Rh under warming conditions may be associated with the rapid decomposition of litter and greater microbial biomass [20]. Wang et al. [21] found in freeze–thaw cycle experiments that temperature is the dominant factor for freeze–thaw occurrence [22]. Soil respiration sharply declines when temperatures are below 0 °C and rapidly increase above 0 °C [23]. A decrease in temperature can affect the quantity and activity of certain soil microorganisms, thereby inhibiting their respiration and leading to reduced soil respiration [24]. This indicates that temperature variation is also crucial for soil respiration and ecosystem responses.

The Tibetan Plateau is a representative region of high-altitude permafrost areas globally [25]. With climate warming, extensive thermal subsidence has formed, and many natural ecological processes on the Tibetan Plateau remain largely intact, providing an opportunity to study environmental changes induced by large-scale climate shifts [26]. Qinghai Lake, located in the northeastern part of the Tibetan Plateau, has its river source wetlands playing a crucial ecological role in responding to climate change and maintaining ecological functions [27]. Under the context of global warming, the temperature conditions, vegetation structure, biological communities, and the physicochemical and microbial characteristics of the soil in the Qinghai Lake river source wetlands and surrounding areas will change [28]. These changes will, in turn, have a profound impact on soil respiration processes and intensity [29], and will significantly influence regional carbon cycling processes and climate change. This study conducted in situ warming and thermal subsidence control experiments at the Wayan Mountain experimental site in the Qinghai Lake area. It analyzes the impacts of these changes on soil respiration and its associated ecological environmental factors, and explores the relationship between soil respiration and soil temperature and moisture. The aim is to provide fundamental data and scientific references for the sustainable development of alpine wetland ecosystems on the Tibetan Plateau.

## 2. Materials and Methods

### 2.1. Experimental Site Description

The study plots are located at the Wayan Mountain experimental research site (37°44′34″ N, 100°5′41″ E) (Figure 1A), which is part of the river source wetlands in Wayan Mountain, Gangcha County, Haibei Tibetan Autonomous Prefecture, in the northeastern part of Qinghai Lake [30] (Figure 1B). The elevation ranges from 3710 m to 3840 m, and the area has a high-altitude continental monsoon climate. The annual average temperature ranges from −3.31 °C to 1.4 °C, and the annual average precipitation is 426.8 mm [30]. The growing season in this study area is from May to September, with August being the peak period for plant growth. The predominant vegetation type is alpine meadow, with the main species being dwarf kobresia (*Kobresia humilis*) and alpine kobresia (*Kobresia pygmaea*), accompanied by other species such as slender sedge (*Carex duriuscula*), lance-leafed Tibetan grass (Tibet Lance), and multi-lobed potentilla (*Potentilla multifida*). The soil type is alpine meadow soil [31]. There is seasonal permafrost, with a maximum depth of about 1.7 m [32]. Below this, there is continuous permafrost, with freezing and thawing periods each lasting half of the year [33]. This region is a typical area for thermokarst development within the Qinghai Lake Basin.

### 2.2. Site Selection and Design

The data for this study were monitored and collected from June to September 2023. Three natural state sites and three thermal subsidence areas (with each subsidence approximately 9 cm deep) were randomly selected to establish control and warming treatments. Four treatments were set up, namely control (CK), warming (W), thermal subsidence (RR), and combined thermal subsidence and warming (RRW), with three replicates for each treatment.

The control group is selected from areas that are non-thermokarst and non-warming. Using systematic random sampling, three thermokarst areas with moderate to severe subsidence were selected. A measuring tape was used to evenly measure 20 times around the edge of the depression in each randomly chosen thermokarst, with an average depth of about 9 cm recorded (Figure 1C). The vegetation primarily consists of Kobresia grass, which is about 40% shorter in height and has a 60% lower cover than the control group. The distance between each thermokarst is approximately 3 to 5 m [31].

An Open-Top Chamber (OTC) was used to simulate warming (Figure 1D), a device widely employed in alpine grassland ecosystems [34]. It is formed by a trapezoid with a base diameter of 2.10 m, a top diameter of 1.50 m, and a height of 0.6 m, with each acrylic panel inclined at an angle of 60° to the ground, creating a regular hexagon. The warming principle relies on the OTC’s heat-trapping effect, which utilizes the penetration of solar infrared radiation through the acrylic to reduce indoor air turbulence and wind speed, thereby achieving a warming effect [32].

The combined treatment of thermal melting subsidence and heating is an experiment conducted in a heat-melting subsidence area using a top-opening growth chamber (Figure 1E).
Figure 1The effects of warming and thermal subsidence on soil respiration in the Qinghai Lake basin experiment. (**A**) Qinghai Lake Basin and the location of the Wayan Mountain experimental research site. (**B**) Experimental sites for warming and thermal subsidence. (**C**) Measuring the soil respiration rate in thermokarst subsidence areas. (**D**) Measuring the soil respiration rate under warming treatment. (**E**) The interaction between OTC warming and thermal subsidence.
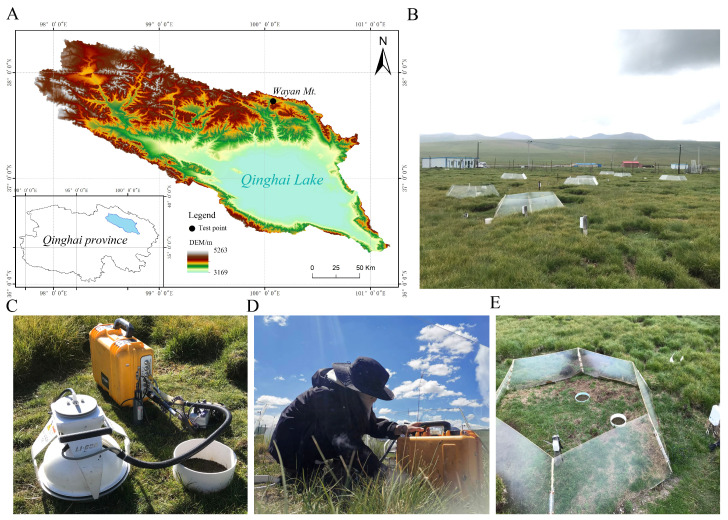


### 2.3. Determination of Soil Ecological and Environmental Factors and Microbial Diversity

Air temperature (AT) and atmospheric relative humidity (RH) meteorological data were measured by a Dynnamet CR 1000 scientific-grade automatic weather station from Dyanmax, with data collected every half hour.

Soil temperature (T) at a depth of 10 cm and soil volumetric water content (VWC) were measured using a portable soil respiration system (Li-8100A, Li-Cor Inc., Lincoln, NE, USA) with its built-in temperature probe and humidity sensor, concurrently with soil respiration measurements. Soil pH was determined by mixing a soil-distilled water suspension (with a mass-to-volume ratio of 1:2.5), shaking it (200 rpm for 60 min), and then measuring with a pH meter. The conductivity was measured using a conductivity meter, with a soil to distilled water mixing ratio of 1:5.

From June to September 2023, a 25 cm × 25 cm quadrat was used to cut the aboveground parts of plants in the study area to ground level. The harvested plants were placed in envelopes and taken back to the laboratory, where they were subjected to a killing treatment in an oven at 105 °C for 30 min. They were then dried at a constant temperature of 65 °C for 24 h until a constant weight was achieved. After weighing, the aboveground biomass was calculated [35].

The underground biomass was obtained using the root coring method. A root corer with a diameter of 10 cm was used to collect soil samples at depths of 0–10 cm and 10–20 cm. The retrieved soil samples were brought back to the laboratory, where a 200-mesh sieve was used to extract the plant roots. The cleaned roots were then placed in a 65 °C constant temperature oven and dried to a constant weight before being weighed, representing the underground biomass [36].

Microbial DNA from soil samples was extracted using the PowerSoil DNA Isolation Kit (Mio-bio, Carlsbad, CA, USA). The quality of the DNA was assessed through agarose gel electrophoresis, and qualified DNA was stored at −80 °C for subsequent PCR amplification. After extracting total genomic DNA from the samples, primers were designed to amplify two hypervariable regions of the bacterial 16S rRNA gene (V3 and V4 regions) and the ITS region (ITS1 and ITS2) for fungi. The forward primers for the bacterial V3 + V4 regions and fungal ITS region are 5′-CCTAYGGGRBGCASCAG-3′ and 5′-CTTGGTCA-TTTAGAGGAAGTAA-3′, respectively. The reverse primers are 5′-GGACTACNNGGGTATCTAAT-3′ and 5′-GCTGCGTTCTTCATCGATGC-3′ [37,38]. PCR amplification was performed in a 50 μL reaction system with the following program: initial denaturation at 98 °C for 1 min, followed by 30 cycles of denaturation at 98 °C for 10 s, annealing at 50 °C for 30 s, and extension at 72 °C for 30 s. A final extension was conducted at 72 °C for 5 min, followed by a hold at 10 °C. After completing the PCR amplification, agarose gel electrophoresis was used to assess the quality of the amplification products [39]. Qualified PCR products were sequenced using high-throughput sequencing, typically employing next-generation sequencing (NGS) technology. After sequencing, bioinformatics tools were used to process the raw data, including filtering low-quality sequences, removing chimeras, and classifying sequences, with Mothur software 1.45.3 commonly used [40]. Diversity indices, such as the Shannon diversity index and Simpson index, were calculated to quantify the diversity of microbial communities.

### 2.4. Soil Respiration Measurement

The trenching method was used to separate the components of soil respiration [41]. Rings were installed one year prior to measurement (May 2022) to allow sufficient time for the remaining plant roots to die. Specifically, PVC soil respiration rings of different depths were buried in each plot. One PVC ring (10 cm high, 20 cm in diameter) was buried 7 cm in the soil to measure total soil respiration, while another PVC ring (50 cm high, 20 cm in diameter) was buried 45 cm deep. Research indicates that nearly 90% of plant roots in the area are distributed in the topsoil (0–30 cm) [42]. The CO_2_ flux measured by the deep ring can represent soil heterotrophic respiration, as the 45 cm PVC ring can sever old plant roots, preventing new roots from growing inside the ring and thereby eliminating root-related respiration. Additionally, to fully block aboveground plants from supplying root carbon within the rings, weeds inside the PVC rings were cut at ground level with scissors the day before measurements were taken. Autotrophic respiration is calculated as the difference between total soil respiration and heterotrophic respiration [43].

During the growing season from June to September 2023, total soil respiration and heterotrophic respiration were measured using a LI-COR 8100 portable soil respiration system (Li-8100A, Li-Cor Inc., Lincoln, NE, USA). Measurements were taken twice a month, conducted between 9:00 AM and 1:00 PM local time.

### 2.5. Data Statistical Analysis

Data preprocessing was conducted using Microsoft Excel 2021. Charts were created with Origin 2021, and statistical analysis was performed using SPSS 26.0. The ACE index, Chao1 index, Shannon index, and Simpson index were used to express bacterial and fungal α-diversity. One-way ANOVA was employed to analyze differences between treatments, and repeated measures ANOVA (with a significance level set at 0.05) was used to assess the effects of warming, thawing treatments, months, and their interactions on soil respiration, autotrophic respiration, heterotrophic respiration, soil volumetric water content, and soil temperature. 

The Q_10_ used in this study was calculated using the van’t Hoff equation (i.e., an exponential function) [44], with the formula as follows:(1)$$R={ae}^{bT}$$
where R represents the respiration rate (μmol m^−2^s^−1^); a, b are the pending parameters, a represents the respiration rate at a temperature of 0 °C, b represents the temperature sensitivity coefficient of respiration; T represents the soil temperature (°C).

The calculation of respiratory temperature sensitivity is as follows:(2)$$Q_{10}=e^{10b}$$
where b is the temperature sensitivity coefficient derived from Equation (1), and Q_10_ represents the temperature sensitivity of respiration, indicating the fold change in soil respiration rate with a 10 °C increase in temperature. This measure assesses the responsiveness of soil respiration to temperature changes [45].

## 3. Results

### 3.1. Air Temperature and Humidity Observations in the Experimental Area for 2023

As shown in Figure 2, the annual average air temperature in the experimental area for 2023 was −2.7 °C, with a monthly maximum of 8.7 °C in August and a minimum of −15.9 °C in January (Figure 1A,C). The annual average relative humidity was 60.7%, with higher monthly relative humidity observed from June to September, recorded at 75.7%, 74.3%, 74.8%, and 76.4%, respectively (Figure 1B,C).

### 3.2. Response of Soil Temperature, Soil Volumetric Water Content, and Biomass to Different Treatments

The changes in soil temperature and soil volumetric water content in the experimental area from June to September 2023 are shown in Figure 2. Under the four treatments, soil temperature exhibited a trend of initially increasing and then decreasing (Figure 3A). The average soil temperatures for W, RR, RRW, and CK treatments were 9.3 °C, 8.3 °C, 9.7 °C, and 8.7 °C, respectively. Notably, the soil temperature in the RRW treatment was significantly higher than that in the W treatment, indicating that the warming treatment had a more pronounced effect during low temperatures. The soil volumetric water content showed a gradual decreasing trend under all four treatments (Figure 3B), with the order of soil volumetric water content being RRW < RR < W < CK. Both the RR and RRW treatments significantly reduced soil volumetric water content (*p* < 0.05), while the W treatment exhibited a decrease, but the difference was not significant (*p* > 0.05). In terms of aboveground biomass (Figure 3C), the W, RR, and RRW treatments significantly reduced aboveground biomass by 48%, 77%, and 58%, respectively, compared to CK (*p* < 0.05). The belowground biomass decreased by 51.94%, 54.08%, and 88.91%, respectively, under the same treatments.

### 3.3. The Impact of Different Treatments on Microbial Diversity

As shown in Table 1, a comparison was made on the effects of control (CK), warming (W), and thermokarst (RR) treatments on soil microbial communities at different soil depths (0–10 cm and 10–20 cm). The results show that in the 0–10 cm soil layer, the bacterial ACE and Chao1 indices under warming treatment (W1) were significantly higher than those in the control (CK1) and thermokarst subsidence (RR1) treatments, indicating that warming significantly promoted bacterial richness. However, in the 10–20 cm soil layer, the ACE and Chao1 indices under warming treatment (W2) were lower, reflecting a reduction in bacterial richness. Regarding fungal communities, in the 0–10 cm soil layer, the ACE index for the control treatment (CK1) was significantly higher than that of other treatments, indicating that warming (W1) and thermokarst subsidence (RR1) reduced fungal richness. In the 10–20 cm soil layer, warming treatment promoted fungal richness, while thermokarst subsidence treatment suppressed it. In terms of evenness, the Simpson index for bacteria was 0.01 across all treatments, suggesting little difference in bacterial community evenness. The Simpson index for fungi was highest under the thermokarst subsidence treatment (RR2), indicating lower evenness in the fungal community under this treatment. The Shannon index for bacteria showed no significant differences among treatments, indicating stable bacterial diversity. In contrast, the Shannon index for fungi was higher under warming (W2) and control (CK2) treatments, demonstrating that these treatments promoted fungal diversity.

### 3.4. The Impact of Different Treatments on Soil Respiration and Its Environmental Factors

As shown in Figure 4, during the 2023 growing season, the seasonal dynamics of soil respiration rate (Rs), soil autotrophic respiration rate (Ra), and soil heterotrophic respiration rate (Rh) exhibited a similar trend, presenting a unimodal curve. The Rs, Ra, and Rh values for the warming (W) and control (CK) treatments reached their peak in July, with the highest values under the W treatment being 8.53 μmol·m^−2^·s^−1^, 4.73 μmol·m^−2^·s^−1^, and 3.80 μmol·m^−2^·s^−1^, respectively. For the CK treatment, the peak values were 11.35 μmol·m^−2^·s^−1^, 8.40 μmol·m^−2^·s^−1^, and 2.95 μmol·m^−2^·s^−1^. The Rs, Ra, and Rh values for the thermokarst (RR) and RR warming (RRW) treatments peaked in August, with the highest values for the RR treatment being 4.83 μmol·m^−2^·s^−1^, 1.58 μmol·m^−2^·s^−1^, and 3.25 μmol·m^−2^·s^−1^, respectively. Under the RRW treatment, the peak values were 2.35 μmol·m^−2^·s^−1^, 0.58 μmol·m^−2^·s^−1^, and 1.77 μmol·m^−2^·s^−1^. Compared to the control, Rs decreased by 11.8%, 73.9%, and 44.7% for W, RR, and RRW, respectively. Ra decreased by 42.3%, 88.6%, and 74.1%, while RR caused Rh to decrease by 44.7%, and W and RRW resulted in increases of 53.5% and 14.4%, respectively. As shown in Table 2, a repeated measures ANOVA indicated that RR, RRW, and the month had significant effects on the soil respiration rate, autotrophic respiration rate, heterotrophic respiration rate, soil temperature, and soil volumetric water content (*p* < 0.05). The interaction between RRW and month significantly affected soil respiration rate and autotrophic respiration rate (*p* < 0.05), but had no significant effect on heterotrophic respiration rate (*p* > 0.05). During the observation period, W had no significant effect on soil respiration rate, autotrophic respiration rate, and heterotrophic respiration rate (*p* > 0.05).

### 3.5. Relationship Between Soil Respiration, Environmental Factors, and Monitoring Indicators

Figure 5 shows that under the four treatments, soil respiration rate and heterotrophic respiration rate are significantly positively correlated with both soil temperature and air temperature. The soil autotrophic respiration rates for CK, W, and RRW also exhibit significant positive correlations with soil temperature and air temperature (*p* < 0.05). Under W and RR treatments, autotrophic respiration rate is positively correlated with soil volumetric water content. Additionally, in the CK treatment, the bacterial ACE index and Shannon index, along with the fungal ACE index, are positively correlated with soil respiration, while the Shannon indices of bacteria and fungi in RR and W treatments show negative correlations with soil respiration rate, but these are not significant (*p* > 0.05). Additionally, soil temperature shows a significant positive correlation with air temperature under all four treatments (*p* < 0.05). In CK and W treatments, soil volumetric water content is significantly positively correlated with electrical conductivity (*p* < 0.05). The bacterial ACE index in CK is positively correlated with pH, soil temperature, air temperature, and aboveground biomass, while soil volumetric water content is significantly positively correlated with underground biomass (*p* < 0.05). Under the W treatment, soil pH shows a significant negative correlation with soil volumetric water content and electrical conductivity, while soil temperature is significantly positively correlated with air temperature (*p* < 0.05). The bacterial ACE index and Shannon index are positively correlated with electrical conductivity, but negatively correlated with soil temperature and air temperature (*p* < 0.05). Under the RR treatment, underground biomass is positively correlated with soil temperature, air temperature, and soil respiration rate (*p* < 0.05). In the RRW treatment, soil pH shows significant positive correlations with soil temperature, soil respiration rate, and its environmental factors, while soil temperature is significantly positively correlated with air temperature (*p* < 0.05).

### 3.6. Effects of Different Treatments on the Temperature Sensitivity of Soil Respiration

As shown in Figure 6 and Table 3, different treatments affect soil respiration rates, autotrophic respiration rates, and heterotrophic respiration rates. For soil respiration rates (Figure 6A), the thawing subsidence (RR) treatment has the highest goodness of fit (R^2^ = 0.88), with a Q_10_ value of 7.39, indicating that soil respiration under RR treatment is most sensitive to temperature changes. In contrast, the warming (W) treatment has the lowest goodness of fit (R^2^ = 0.23) and a lower Q_10_ value (3.00), suggesting a weaker response to temperature variations. For soil autotrophic respiration rates (Figure 6B), the control (CK) has the highest goodness of fit (R^2^ = 0.19) and a Q_10_ value of 4.95, indicating a significant response to temperature changes, while the Q_10_ value for the RR treatment is lower (2.23). Regarding soil heterotrophic respiration rates (Figure 6C), both the RR and thawing subsidence (RRW) treatments show a high goodness of fit (0.73 and 0.65) and higher Q_10_ values (12.18 and 4.95), suggesting that heterotrophic respiration in these treatments is more sensitive to temperature variations. In comparison, the control (CK) and warming (W) treatments have lower goodness of fit (0.09 and 0.21) and Q_10_ values of 1.65 and 2.23, respectively, indicating that heterotrophic respiration under these treatments responds weakly to temperature changes. Overall, the effects of different treatments on soil respiration rates are significantly distinct, particularly regarding the temperature sensitivity of autotrophic and heterotrophic respiration. The warming and thawing subsidence treatments show clear differences in their temperature responses for soil respiration.

## 4. Discussion

### 4.1. The Effects of Different Treatments on Soil Respiration Rates and Their Environmental Factors

Research findings on the impact of warming on soil respiration and its environmental factors remain controversial in the context of global climate change [46]. Previous studies have shown that warming generally increases soil respiration rates [19], while some researchers argue that warming can decrease soil respiration rates [47]. This study found that under warming conditions, soil respiration rates and autotrophic respiration rates weakened, while heterotrophic respiration rates increased. However, the overall impact of warming on soil respiration was not significant (*p* > 0.05). This may be due to long-term warming enhancing the adaptability of soil microorganisms to temperature [48], leading to an overall declining trend in soil respiration rates [49,50]. For deep soil, warming can enhance the activity of soil microorganisms and enzymes, stimulating soil heterotrophic respiration. The increase in soil enzyme activity is associated with the rise in soil microorganism numbers and microbial activity under warming conditions. The increase in soil microorganism numbers and the enhancement of microbial activity stimulate microbial metabolic activity, accelerating soil carbon mineralization and promoting soil heterotrophic respiration [51,52]. The decrease in autotrophic respiration rates may be related to changes in the annual averages of fine root production and mortality during warming experiments [53].

In this study, the monthly variation of soil respiration rate in the thermal melting subsidence area exhibited a unimodal curve. Compared to the control group, the soil respiration rate in the thermal melting subsidence area reached its peak in August. This is primarily due to the slower temperature increase in the subsidence area, with August being the peak growth season for plants. The developed root systems of the plants exhibit strong respiration, and the microbial activity in the soil also reaches its most active state of the year, thereby enhancing the soil respiration rate. However, compared to the natural state, the soil respiration rate decreased by 73.9% (*p* < 0.05), the autotrophic respiration rate decreased by 88.6% (*p* < 0.05), and the heterotrophic respiration rate decreased by 44.7% (*p* < 0.05). This result is consistent with findings by Ding et al. [31], which showed that soil respiration rates in freeze–thaw erosion areas were lower than those in the natural state. This may be related to the low plant biomass and small seasonal variation in the thermal melting subsidence area, where the seasonal variation of root respiration is less than that in the natural state [54]. Additionally, it may also be related to the duration of thermal melting subsidence. Long-term thermal melting subsidence could lead to the destruction or removal of surface vegetation and the soil organic matter layer, resulting in significant losses of carbon and nutrients [55]. Organic matter may also be transported to downstream ecosystems through processes such as mudflows, sliding, and collapse [56], or it may infiltrate deeper into the soil [57].

Climate warming has different effects on ecological processes in thermokarst and non-thermokarst regions. In thermokarst areas, the OTC (open-top chamber) devices have resulted in an average soil temperature increase of 1.4 °C for depths greater than 10 cm. The warming treatment has led to an average increase of 52.8% in soil respiration (Rs). This is consistent with the findings of Wang et al. [41]. In thermokarst regions, warming exacerbates soil CO_2_ loss, and the warming effect is amplified compared to non-thermokarst areas, reinforcing the positive feedback between permafrost carbon and climate [41]. This effect may be related to the significant decline in organic carbon quality and soil microbial diversity in thermokarst areas, making soil matrix decomposition and microbial-mediated functions more sensitive to temperature increases [58,59].

### 4.2. The Impact of Environmental Factors on Soil Respiration and Their Environmental Influences

In the process of global climate change, changes in temperature and precipitation patterns occur simultaneously [60], inevitably leading to alterations in abiotic factors (moisture, temperature, pH) and biotic factors (microbial diversity, aboveground and belowground biomass). These changes directly or indirectly affect soil respiration and the response of its environmental factors to different treatments (CK, W, RR, RRW) [61]. The results of this study indicate that soil respiration shows a positive correlation with near-surface air temperature and soil temperature under different treatments. This is consistent with the findings of Wang et al. The main reason for this is that the increase in soil temperature directly affects plant root activity and soil microbial activity, thereby promoting the decomposition of organic matter and the release of CO_2_ [62]. Compared to the control group, both warming and thermokarst reduced soil volumetric water content. Under warming treatment, soil respiration rates decreased, while soil autotrophic respiration showed a positive correlation with soil volumetric water content. In thermokarst areas, there was no significant correlation between soil respiration processes and soil volumetric water content (Figure 4). Yu et al. demonstrated in warming simulation experiments in alpine meadows that as soil moisture decreases, wet meadows transition to dry meadows, leading to a weakened impact of warming on soil respiration, which aligns with the warming results in this study [63]. In this study, the correlation between soil temperature and soil respiration was consistently stronger than that between soil volumetric water content and soil respiration under different treatments. Lin et al. also indicated a positive correlation between respiration and temperature, with a less significant relationship to soil moisture, aligning with our findings [64]. This is likely related to the high alpine marsh wetland context, where warming and thermokarst treatments had minimal effects on soil volumetric water content. Soil temperature and moisture can directly affect the growth environment of plants, while changes in plant biomass can alter the amount of residue and litter in the soil. These changes can further influence the activity and community structure of soil microorganisms, thereby indirectly affecting the intensity of soil respiration [65]. The results of this study indicate that aboveground biomass under warming conditions decreased by 48.2% compared to natural conditions. Research by Zhao et al. also showed that long-term warming leads to a reduction in aboveground biomass, which aligns with our findings [66]. This reduction may be due to the OTC warming devices lowering wind speed, increasing litter biomass, and reducing light availability [34]. In contrast, the total underground biomass under warming conditions was higher than in natural states, likely because the experimental site is located in a river source wetland with sufficient moisture, allowing warming to significantly enhance underground biomass. Under thermokarst conditions, both aboveground and underground biomass significantly decreased, which is consistent with the findings of Ding Junxia et al. This reduction may be related to the decreased soil organic matter content in thermokarst areas, leading to insufficient nutrient supply for plants and affecting both aboveground and underground biomass [31,67]. Under the interaction of thermokarst and warming, biomass decreased compared to the control conditions, but significantly increased relative to the thermokarst state. This suggests that warming may mitigate the adverse effects of thermokarst through an “stimulating” effect.

Microorganisms are a vital component of soil ecosystems and are highly sensitive to environmental changes; both warming and thermokarst can affect microbial activity [68]. Soil properties are fundamental environmental conditions for microbial survival. This study demonstrates the impact of soil properties (such as temperature, moisture, pH, and EC) on the α-diversity of bacteria and fungi. The results show that the ACE index of bacteria under CK and W treatments strongly correlates with soil properties, consistent with Zhang et al.’s findings that soil physicochemical properties are key limiting factors controlling changes in microbial α-diversity [69]. Under the RR treatment, the ACE index of fungi exhibited a negative correlation with soil volumetric water content, likely due to the varying responses of microbial communities to environmental changes across different treatments. Under warming and thermokarst treatments, microbial α-diversity increased. Existing studies have indicated that the Shannon and Simpson indices of soil bacteria and fungi correlate with soil respiration rates, with respiration rates increasing alongside bacterial and fungal diversity [69]. In this study, the soil respiration rate under CK showed a positive correlation with the Shannon index of soil bacteria, while the respiration rates under RR and W exhibited a negative correlation with both bacterial and fungal Shannon indices. This may be attributed to long-term warming reducing microbial biomass and the availability of substrates [70,71], which directly impacts microbial diversity and leads to a decrease in soil respiration rates and temperature sensitivity, indicating a stronger temperature adaptability. Compared to the control group, soils in thermokarst areas experience repeated freeze–thaw cycles, leading to the increased mortality of soil microorganisms. This results in decreased microbial activity and biomass, ultimately causing a reduction in soil respiration rates [72,73,74].

Soil pH is an important factor influencing soil respiration in ecosystems [75]. In soils with a pH less than 7, as pH increases, the soil respiration rate also increases [76]. This study found that under natural conditions, soil respiration is positively correlated with pH. However, under warming, thermal melting subsidence, and their interaction, the soil becomes slightly alkaline, where the interaction between thermal melting subsidence and these treatments shows a positive correlation between pH and soil respiration. This may be due to pH affecting microbial metabolic responses, as microorganisms involved in organic carbon decomposition and nitrogen nitrification processes are highly sensitive to pH [77]. Conversely, under warming conditions, pH shows a negative correlation with soil respiration, possibly because salinity inhibits the release of soil CO_2_ into the atmosphere [78].

## 5. Conclusions

This study provides important insights into the effects of warming and thermokarst on soil properties and processes in the river source wetland of the Qinghai Lake basin. The main conclusions are as follows: 1. The effects of warming and thermokarst on soil temperature and moisture: Warming treatments significantly increased soil temperature, particularly noticeable during colder seasons, while thermokarst exacerbated this effect. Thermokarst also led to a significant reduction in soil volumetric water content, with the RRW treatment having the most pronounced impact on moisture. 2. Changes in biomass: compared to the control group, warming and thermokarst treatments significantly reduced both aboveground and underground biomass, indicating that these environmental changes have a substantial impact on vegetation productivity. 3. Response of microbial diversity: Warming treatments promoted the richness of bacteria in surface soils, while thermokarst inhibited the diversity of fungal communities. These changes in microbial diversity reflect the differing impacts of environmental stress on microbial communities. 4. Dynamics of soil respiration: Soil respiration rates exhibited a unimodal curve during the growing season. Warming significantly reduced autotrophic respiration rates, while thermokarst suppressed heterotrophic respiration. These responses indicate the complex interactions between soil temperature, moisture, and microbial activity. 5. Temperature sensitivity of soil respiration: The thermokarst treatment exhibited the highest sensitivity to temperature changes, with a Q_10_ value as high as 7.39, indicating a strong response to climate warming. This suggests that changes in soil conditions induced by thermokarst may amplify soil carbon emissions.

This study highlights the profound impacts of warming and thermokarst induced by climate change on the carbon cycling of alpine wetland soils. These findings provide important scientific evidence for understanding the response mechanisms of soil carbon cycling to climate change.

## Figures and Tables

**Figure 2 biology-13-00863-f002:**
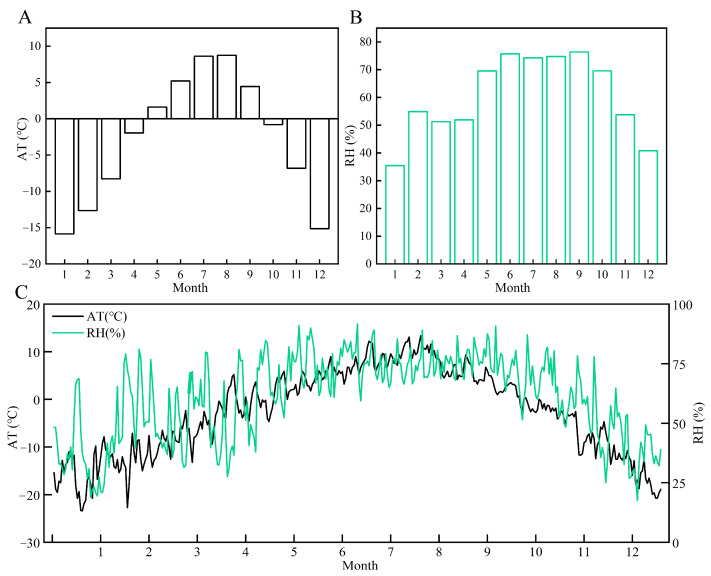
The monthly average temperature and humidity, and the variation curves of air temperature and humidity in the experimental area for 2023. (**A**) Monthly average temperature for 2023. (**B**) Monthly average humidity for 2023. (**C**) Variation curve of air temperature and humidity in the experimental area for 2023.

**Figure 3 biology-13-00863-f003:**
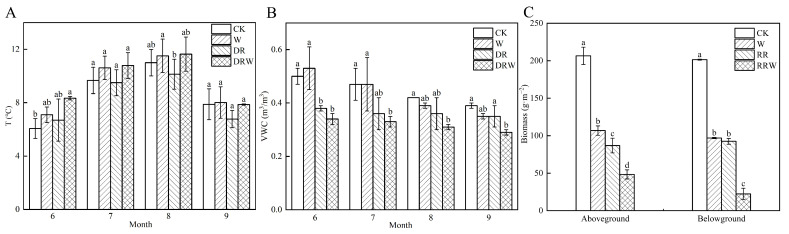
Changes in soil temperature, volumetric water content, and aboveground and belowground biomass under different treatments. (**A**) Soil temperature variation from June to September 2023; (**B**) Soil volumetric water content variation from June to September 2023; (**C**) Changes in aboveground and belowground biomass. Note: CK: control; W: warming treatment; RR: thaw settlement area; RRW: warming treatment in thaw settlement area. Different letters indicate significant differences between treatments (*p* < 0.05).

**Figure 4 biology-13-00863-f004:**
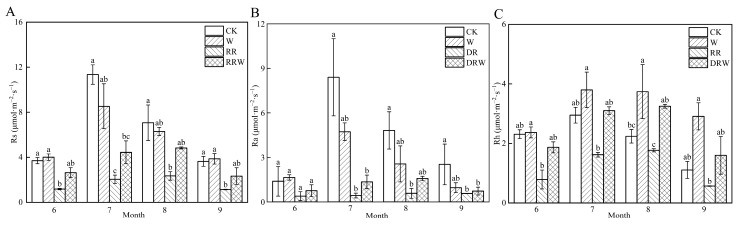
Effects of warming and thermokarst on soil respiration rate. (**A**) Soil respiration rate; (**B**) autotrophic respiration rate; (**C**) heterotrophic respiration rate. Note: CK: control; W: warming; RR: thermokarst; RRW: warming treatment of thermokarst. Different letters indicate significant differences between treatments (*p* < 0.05).

**Figure 5 biology-13-00863-f005:**
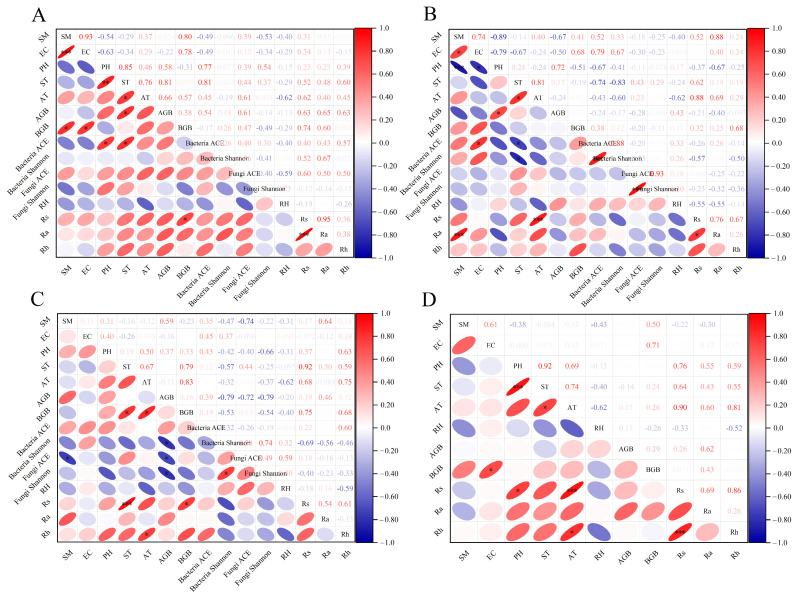
The correlations between soil respiration and its environmental factors across the four treatments. (**A**) Control treatment; (**B**) warming treatment; (**C**) thermokarst treatment; (**D**) combined thermokarst and warming treatment. Note: VWC represents soil volumetric water content, EC represents soil electrical conductivity, pH represents acidity, ST represents soil temperature, AT represents air temperature, AGB represents aboveground biomass, BGB represents underground biomass, RH represents air relative humidity, RS represents soil respiration rate, Ra represents autotrophic respiration rate, and Rh represents heterotrophic respiration rate. * indicates *p* < 0.05; ** indicates *p* < 0.01; *** indicates *p* < 0.001.

**Figure 6 biology-13-00863-f006:**
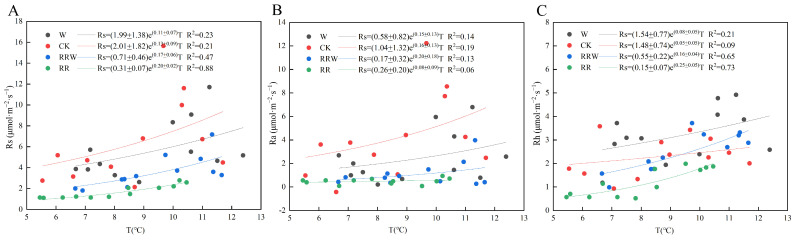
Nonlinear fitting of soil respiration rates and their environmental factors with soil temperature under different treatments; (**A**) soil respiration rate; (**B**) soil autotrophic respiration rate; (**C**) soil heterotrophic respiration rate.

**Table 1 biology-13-00863-t001:** Soil microbial diversity indices under warming and thermokarst treatments.

Groups		RR1	W1	CK1	RR2	W2	CK2
ACE	Bacteria	1424.00 ± 5.96 b	1498.99 ± 137.20 a	1387.75 ± 14.38 c	1422.87 ± 7.49 b	1339.17 ± 11.12 d	1505.03 ± 14.11 a
	Fungi	52.46 ± 6.92 b	58.54 ± 24.82 ab	69.30 ± 1.54 ab	49.37 ± 15.18 b	77.49 ± 2.27 a	66.46 ± 6.96 ab
Chao1	Bacteria	1431.97 ± 3.86 b	1505.18 ± 135.12 a	1399.54 ± 10.83 c	1439.66 ± 4.87 b	1354.68 ± 9.60 d	1516.35 ± 18.54 a
	Fungi	50.39 ± 6.23 c	59.43 ± 24.55 abc	69.92 ± 1.77 ab	51.12 ± 13.57 bc	76.12 ± 7.63 a	66.73 ± 8.49 abc
Simpson	Bacteria	0.01 ± 0.00 a	0.01 ± 0.00 a	0.01 ± 0.00 a	0.01 ± 0.00 a	0.01 ± 0.00 a	0.06 ± 0.10 a
	Fungi	0.30 ± 0.13 a	0.24 ± 0.07 a	0.30 ± 0.19 a	0.46 ± 0.38 a	0.16 ± 0.01 a	0.16 ± 0.05 a
Shannon	Bacteria	6.19 ± 0.03 a	6.20 ± 0.05 a	6.00 ± 0.05 a	5.82 ± 0.09 a	5.98 ± 0.03 a	5.34 ± 1.33 a
	Fungi	1.90 ± 0.47 a	2.04 ± 0.35 a	2.14 ± 0.70 a	1.52 ± 1.05 a	2.54 ± 0.12 a	2.54 ± 0.23 a

Note: RR1, W1, and CK1 represent the 0–10 cm soil layer, while RR2, W2, and CK2 represent the 10–20 cm soil layer. Different letters indicate significant differences between treatments (*p* < 0.05).

**Table 2 biology-13-00863-t002:** Repeated measures ANOVA (F values) for soil respiration rate, soil temperature, and soil volumetric water content under different treatments.

Variable	Warming	Thermokarst	Month	Warming × Thermokarst	Warming × Thermokarst × Month
Soil respiration (Rs) (μmol·m^−2^·s^−1^)	0.229	15.368 *	23.059 ***	6.580 *	3.477 **
Autotrophic respiration (Ra) (μmol·m^−2^·s^−1^)	1.839	12.896 *	9.723 ***	5.621 *	3.267 *
Heterotrophic respiration (Rh) (μmol·m^−2^·s^−1^)	2.064	170.893 ***	13.902 ***	7.030 *	1.025
Soil temperature (T) (°C)	12.281	0.072	21.934 ***	2.0336	0.234
Soil volumetric water content (VWC) (m^3^/m^3^)	0.127	19.512 *	8.648 *	9.963 **	1.115
Aboveground biomass (g/m^2^)	127.385 ***	87.687 **	27.334 ***	37.023 ***	7.051 ***
Root biomass at 0–10 cm (g/m^2^)	64,895.21 ***	0.213	27.745 ***	6.33 *	5.015 **
Root biomass at 10–20 cm (g/m^2^)	78.315 **	2126.094 ***	98.538 ***	111.436 ***	13.19 ***

Note: * indicates *p* < 0.05; ** indicates *p* < 0.01; *** indicates *p* < 0.001.

**Table 3 biology-13-00863-t003:** Soil respiration rate models and indicators under different treatments.

Treatment	Soil Respiration Rate			Soil Autotrophic Respiration Rate			Soil Heterotrophic Respiration Rate		
	Exponential Equation	R^2^	Q_10_	Exponential Equation	R^2^	Q_10_	Exponential Equation	R^2^	Q_10_
W	y = 1.99e^0.11T^	0.23 ***	3.00	y = 0.58e^0.15T^	0.14 **	4.48	y = 1.54e^0.08T^	0.21 ***	2.23
CK	y = 2.01e^0.13T^	0.21 ***	3.67	y = 1.04e^0.16T^	0.19 **	4.95	y = 1.48e^0.05T^	0.09 ***	1.65
RRW	y = 0.71e^0.17T^	0.47 ***	5.47	y = 0.17e^0.20T^	0.13 **	7.39	y = 0.55e^0.16T^	0.65 ***	4.95
RR	y = 0.31e^0.20T^	0.88 ***	7.39	y = 0.26e^0.08T^	0.06 ***	2.23	y = 0.15e^0.25T^	0.73 ***	12.18

Note: ** indicates *p* < 0.01; *** indicates *p* < 0.001.

## Data Availability

All data generated or analyzed during this study are included in this published article.

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
