# Peer review of "Responses of Soil Respiration and Ecological Environmental Factors to Warming and Thermokarst in River Source Wetlands of the Qinghai Lake Basin"

_biology, 2024, doi:10.3390/biology13110863_

Round 1

Reviewer 1 Report

Comments and Suggestions for Authors

The manuscript investigated the effects of warming and thermokarst on soil respiration and environmental factors, by using Open Top Chamber device to simulate warming on natural and thermal subsidence areas. Generally, the ms is well-written and the major conclusions are clearly presented. However, the methodology, especially for the experimental design, needs add more details to get a solid conclusion.

Here are my concerns:

1.      In the method section, there should be adjacent sampling areas with same environmental settings except of your treatment, i.e., warming enhancing device—OTC, as well as the thaw settlement. However, I didn’t see any environmental descriptions of the sampling sites, so as I cannot figure out the differences (such as soil characteristics) between those sampling areas, before your treatment setup.

2.      Line 87, please define Rh during its first mention.

3.      Line 91-92, any links with the previous sentence? Additionally, full name are not required for the citations, check the citations throughout the ms.

4.      Line 118, check the figure label.

5.      Figure 1, cannot see the sampling sites, i.e., natural areas and thermokarst areas.

6.      Line 169, specify whether the subsamples used for molecular analysis were extracted immediately or provide details on the transfer and storage procedures.

7.      Line 187-191, it is expected that references for your soil respiration measurements, particularly for the calculation of autotrophic respiration, should be provided.

8.      Line 216-218, I cannot see it clearly from the figure. Is it statistically significant?

9.      Line 254-258, rephrase the sentence.

10.   Line 258-268, simplify the presentation of the results to clearly outline your major findings

11.   Table 1, bacteria and fungi are not microbial species.

12.   Line 313, include the measurement method of electrical conductivity in the methods section.

13.   Line 370-373, the statements here cannot explain your major findings, that soil respiration and autotrophic respiration rates were weakened, while heterotrophic respiration rates increased under warming conditions.

14.   Line 390-399, the comparison of warming effects on soil ecological processes between thermokarst and non-thermokarst regions is interesting, and should be highlighted in the results and discussion section from my view.

15.   Line 403, microbial diversity cannot be classified as abiotic factors.

16.   Line 405, specify what is meant by 'different treatments'

17.   Line 432, ‘This reduction may be due to the OTC warming devices lowering wind speed, increasing litter biomass, and reducing light availability’. That is exactly what I am concerned. How do you control the variables to ensure that environmental factors other than your treatment remain consistent?

18.   Line 471, clarify why salinity is attributed to the negative correlations between pH and soil respiration, providing a scientific explanation for this relationship.

Comments on the Quality of English Language

 Minor editing of English language required.

Author Response

Response to reviewer 1

Comments 1: In the method section, there should be adjacent sampling areas with same environmental settings except of your treatment, i.e., warming enhancing device—OTC, as well as the thaw settlement. However, I didn’t see any environmental descriptions of the sampling sites, so as I cannot figure out the differences (such as soil characteristics) between those sampling areas, before your treatment setup.

Response 1: Thank you for your comment. In "2 Materials and Methods," the overview of the study area and the selection and design of sampling plots are described. All four treatments are located in the Wayan Mountain experimental plots, where the soil type is alpine meadow soil. The vegetation types for the control group and warming treatment are dwarf reed grass and alpine reed grass, while the vegetation type in the thermokarst area is small reed grass, which is approximately 40% shorter than that of the control group.

Comments 2: Line 87, please define Rh during its first mention.

Response 2: Thank you for your comment. In lines 80-83 of the introduction, definitions of autotrophic and heterotrophic respiration should be added: Soil respiration is classified into autotrophic and heterotrophic respiration based on the source of CO2. Autotrophic respiration mainly involves CO2 produced by plant root respiration and rhizosphere respiration, while heterotrophic respiration refers to CO2 generated from microbial respiration and soil animal respiration.

Comments 3: Line 91-92, any links with the previous sentence? Additionally, full name are not required for the citations, check the citations throughout the ms.

Response 3: Thank you for your comment. In lines 93-99 of the introduction, the sentence order has been modified to: "Wang et al. [21] found in freeze-thaw cycle experiments that temperature is the dominant factor for freeze-thaw events [22]. Soil respiration sharply declines when temperatures are below 0°C and rapidly increases when above 0 ℃ [23]. A decrease in temperature affects the quantity and activity of certain soil microorganisms, thereby inhibiting their respiration and leading to a reduction in soil respiration [24]. This indicates that temperature changes significantly impact soil respiration and ecosystem responses."

Comments 4:  Line 118, check the figure label.

Response 4: Thank you for your comment. In line 138, modify Figure 1A's caption to "Location of the Qinghai Lake Basin and Wayan Mountain experimental sites." Add Figure 1C in line 153, Figure 1D in line 157, and Figure 1E in line 166.

Comments 5: Figure 1, cannot see the sampling sites, i.e., natural areas and thermokarst areas.

Response 5: Thank you for your comment. We apologize that the image does not clearly display the details due to the distance and angle of the shot. The following image shows that the red box marks the respiration ring in the thermokarst area, while the blue box indicates the respiration ring in natural conditions, although it is partially obscured by taller vegetation.(You can view the image in the attached file)

Comments 6: Line 169, specify whether the subsamples used for molecular analysis were extracted immediately or provide details on the transfer and storage procedures.

Response 6: Thank you for your comment. In section 2.3, revise the sampling of belowground biomass to: "The collected soil samples were brought back to the laboratory, where a 200-mesh sieve was used to extract plant roots. The cleaned roots were placed in a 65°C constant-temperature oven and dried to a constant weight, then weighed to represent belowground biomass."

Comments 7: Line 187-191, it is expected that references for your soil respiration measurements, particularly for the calculation of autotrophic respiration, should be provided.

Response 9: Thank you for your comment. Line 288-304,Rephrase section 3.3 on microbiology.

The results show that in the 0-10 cm soil layer, the bacterial ACE and Chao1 indices under warming treatment (W1) were significantly higher than those in the control (CK1) and thermokarst subsidence (RR1) treatments, indicating that warming significantly promoted bacterial richness. However, in the 10-20 cm soil layer, the ACE and Chao1 indices under warming treatment (W2) were lower, reflecting a reduction in bacterial richness.

Comments 10: Line 258-268, simplify the presentation of the results to clearly outline your major findings.

Response 10: Thank you for your comment. Line 288-304,Rephrase section 3.3 on microbiology.

Regarding fungal communities, in the 0-10 cm soil layer, the ACE index for the control treatment (CK1) was significantly higher than that of other treatments, indicating that warming (W1) and thermokarst subsidence (RR1) reduced fungal richness. In the 10-20 cm soil layer, warming treatment promoted fungal richness, while thermokarst subsidence treatment suppressed it. In terms of evenness, the Simpson index for bacteria was 0.01 across all treatments, suggesting little difference in bacterial community evenness. The Simpson index for fungi was highest under the thermokarst subsidence treatment (RR2), indicating lower evenness in the fungal community under this treatment. The Shannon index for bacteria showed no significant differences among treatments, indicating stable bacterial diversity. In contrast, the Shannon index for fungi was higher under warming (W2) and control (CK2) treatments, demonstrating that these treatments promoted fungal diversity.

Comments 11: Table 1, bacteria and fungi are not microbial species.

Response 11: Thank you for your comment. Change "Species" to "Groups" in Table 1 of section 3.3.

Comments 12: Line 313, include the measurement method of electrical conductivity in the methods section.

Response 12: Thank you for your comment. Line 176-177,add the method for measuring conductivity: Conductivity is measured using a conductivity meter, with a soil to distilled water mixing ratio of 1:5.

Comments 13: Line 370-373, the statements here cannot explain your major findings, that soil respiration and autotrophic respiration rates were weakened, while heterotrophic respiration rates increased under warming conditions.

Response 13: Thank you for your comment. Line 401-415, rewrite the warming section in 4.1.

Previous studies have shown that warming generally increases soil respiration rates [19], while some researchers argue that warming can decrease soil respiration rates [47]. This study found that under warming conditions, soil respiration rates and autotrophic respiration rates weakened, while heterotrophic respiration rates increased. However, the overall impact of warming on soil respiration was not significant (P > 0.05). This may be due to long-term warming enhancing the adaptability of soil microorganisms to temperature [48], leading to an overall declining trend in soil respiration rates [49, 50]. For deep soil, warming can enhance the activity of soil microorganisms and enzymes, stimulating soil heterotrophic respiration. The increase in soil enzyme activity is associated with the rise in soil microorganism numbers and microbial activity under warming conditions. The increase in soil microorganism numbers and the enhancement of microbial activity stimulate microbial metabolic activity, accelerating soil carbon mineralization and promoting soil heterotrophic respiration [51, 52]. The decrease in autotrophic respiration rates may be related to changes in the annual averages of fine root production and mortality during warming experiments [53].

Comments 14: Line 390-399, the comparison of warming effects on soil ecological processes between thermokarst and non-thermokarst regions is interesting, and should be highlighted in the results and discussion section from my view.

Response 14: Thank you for your comment. Line 416-434, rewrite the section on thermokarst subsidence in 4.1.

In this study, the monthly variation of soil respiration rate in the thermal melting subsidence area exhibited a unimodal curve. Compared to the control group, the soil respiration rate in the thermal melting subsidence area reached its peak in August. This is primarily due to the slower temperature increase in the subsidence area, with August being the peak growth season for plants. The developed root systems of the plants exhibit strong respiration, and the microbial activity in the soil also reaches its most active state of the year, thereby enhancing the soil respiration rate. However, compared to the natural state, the soil respiration rate decreased by 73.9% (P<0.05), the autotrophic respiration rate decreased by 88.6% (P<0.05), and the heterotrophic respiration rate decreased by 44.7% (P<0.05). This result is consistent with findings by Ding et al. [31], which showed that soil respiration rates in freeze-thaw erosion areas were lower than those in the natural state. This may be related to the low plant biomass and small seasonal variation in the thermal melting subsidence area, where the seasonal variation of root respiration is less than that in the natural state [54]. Additionally, it may also be related to the duration of thermal melting subsidence. Long-term thermal melting subsidence could lead to the destruction or removal of surface vegetation and the soil organic matter layer, resulting in significant losses of carbon and nutrients [55]. Organic matter may also be transported to downstream ecosystems through processes such as mudflows, sliding, and collapse [56], or it may infiltrate deeper into the soil [57].

Comments 15: Line 403, microbial diversity cannot be classified as abiotic factors.

Response 15: Thank you for your comment. Line 447-450, revise the original sentence to: "This inevitably leads to changes in abiotic factors (moisture, temperature, pH) and biotic factors (microbial diversity, aboveground and belowground biomass), thereby directly or indirectly affecting soil respiration and its response to environmental factors under different treatments."

Comments 16: Line 405, specify what is meant by 'different treatments.

Response 16: Thank you for your comment. Line 451, different treatments refer to the four experimental treatments, including natural conditions, warming, thermokarst subsidence, and the interaction treatment of warming and thermokarst subsidence (CK, W, RR, RRW).

Comments 17: Line 432, ‘This reduction may be due to the OTC warming devices lowering wind speed, increasing litter biomass, and reducing light availability’. That is exactly what I am concerned. How do you control the variables to ensure that environmental factors other than your treatment remain consistent?

Response 17: Thank you for your comment. The open-top chamber (OTC) is widely recognized as a standard method for simulating warming in ecological studies. It has been extensively used in research across various ecosystems, including cold and wetland environments, to ensure consistent temperature increases. While OTCs can slightly alter environmental factors like wind speed and light, these effects are well-documented, and the design allows for careful control and monitoring of variables. This ensures the reliability of the warming treatment, while minimizing unintended environmental variations. The detailed specifications of the OTC setup, including its heating mechanisms, were carefully followed to maintain experimental consistency.

In their 2011 paper published in Global Change Biology, Godfree et al. stated that OTC materials must have good light transmission to avoid damage to photosynthetically active radiation (PAR). Polycarbonate (transmittance 89%), PVC, acrylic, and glass are viable options that maintain PAR above 80%.

In their 2017 article in Annals of Botany, Schollert et al. noted that when the sun is at a low angle, the reduction in photosynthetic photon flux density inside the OTC can reach 16%-25%. While the reduction is greatest, the open top allows for direct sunlight, with reductions approaching zero around noon, especially at lower latitudes.

In their 2022 review published in Arctic Science, Hollister et al. indicated that hexagonal OTC setups are designed to maximize temperature, capture longwave radiation, and minimize wind advection. The shape aims to significantly reduce wind speed and minimize energy loss caused by air movement (advection).

Comments 18: Line 471, clarify why salinity is attributed to the negative correlations between pH and soil respiration, providing a scientific explanation for this relationship.

Response 18: Thank you for your comment. In the study by Wang et al., it was found that soil respiration rate is significantly negatively correlated with exchangeable sodium and pH. As exchangeable sodium and pH increase, the soil respiration rate decreases, meaning that the more severe the salinization of the soil, the less CO2 it releases to the atmosphere through respiration. Duan et al. indicated that, generally, salinity is positively correlated with pH, and higher soil salinity directly affects the quantity and activity of microorganisms in the soil. Zhang et al. found that as soil pH and conductivity increase, the numbers of bacteria, fungi, and actinomycetes in soils with different salinity levels significantly decrease, thereby affecting the soil respiration rate.

References:1.Wang M, Liu X, Li X, et al: Soil respiration dynamics and its controlling factors of typical vegetation communities on meadow steppes in the western Songnen Plain. Chinese Journal of Applied Ecology, 2014, 25(1): 45-52. DOI:10.13287/j.1001-9332.2014.01.007.

2.Duan M Q,Ding X D,Li S M, et al: Spatio-temporal Variability of Soil pH in Typical Areas of the Yellow River Delta.Journal of Irrigation and Drainage, 2020,39(S2):9-13.DOI:10.13522/j.cnki.ggps.2020413.

3.Zhang W,Feng Y J.Distributionofsoil microorganism andtheir relations with soil factors ofsaline-alkaline grasslands inSongnenPlain.Grassland and Turf,2008,(03):7-11.DOI:10.13817/j.cnki.cyycp.2008.03.013.

Reviewer 2 Report

Comments and Suggestions for Authors

I have thoroughly reviewed the manuscript “Responses of Soil Respiration and Ecological Environmental Factors to Warming and Thermokarst in River Source Wetlands of the Qinghai Lake Basin”. The authors have addressed an important topic – effect of global warming on the soils underlined by permafrost. But from my point of view, there are some discrepancies in methodology and results interpreting in the manuscript.

For instance:

Lines 165-166 “The extracted soil was washed with clean water to remove live roots." Please explain the methodology and reason for live root removal.

In the section “2.3. Determination of soil ecological and environmental factors and microbial diversity,” only the DNA extraction procedure is described. Please describe the methodology of microbial diversity determination and its use in the following results interpretation.

From the section “2.4. Soil respiration measurement ” it is not clear what the difference is between “total soil respiration” and “soil heterotrophic respiration” and why the authors consider this difference only in different depths of collars digging. References?

In the section “2.5. Data statistical analysis”:

Line 206: “a represents the respiration rate at temperature 0℃”. How this value measured? If it is not measured, from where did the authors take it?

Lines 210-211: “b is the respiratory temperature sensitivity coefficient derived from formula (1)." Please describe the formula.

In addition, there are some remarks to figures and tables:

Tables 1 and 2 are hard to read.

In Figure 1, the word “measurement” seems missing in “(C) Soil respiration rate in the thermal subsidence area. (D) 132 Soil respiration rate under warming treatment”.

In Figure 4, the error bars visually do not correspond to the letters that indicate significant differences between treatments.

Throughout the text, the control variants of the experiments had their own dynamics, which the authors did not take into account when describing the results. In this regard, I believe that the methodology of the article requires a more complete description and the results a complete revision.

Author Response

Comments 1: Lines 165-166 “The extracted soil was washed with clean water to remove live roots." Please explain the methodology and reason for live root removal.

Response 1: Thank you for your comment. Line 184-189, the original content contained abbreviations and misunderstandings, which have been revised to: "The underground biomass was obtained using a root auger. A root auger with a diameter of 10 cm was used to collect soil at depths of 0-10 cm and 10-20 cm. The collected soil samples were brought back to the laboratory, where a 200-mesh sieve was used to separate the plant roots. The cleaned roots were then placed in a constant temperature oven at 65°C and dried to a constant weight, after which they were weighed to represent the underground biomass [36]."

Comments 2: In the section “2.3. Determination of soil ecological and environmental factors and microbial diversity,” only the DNA extraction procedure is described. Please describe the methodology of microbial diversity determination and its use in the following results interpretation.

Response 2: Thank you for your comment. Line 204-210, add the following content to section 2.3 on microbial assessment: “Qualified PCR products were sequenced using high-throughput sequencing, typically employing next-generation sequencing (NGS) technology. After sequencing, bioinformatics tools were used to process the raw data, including filtering low-quality sequences, removing chimeras, and classifying sequences, commonly using software such as Mothur [40]. Diversity metrics, such as the Shannon diversity index and Simpson index, were calculated to quantify microbial community diversity.”

Line 229, add the following application to section 2.5: “The ACE index, Chao1 index, Shannon index, and Simpson index were used to express bacterial and fungal α-diversity, and one-way ANOVA was conducted to analyze the differences between different treatments.”

Comments 3: From the section “2.4. Soil respiration measurement ” it is not clear what the difference is between “total soil respiration” and “soil heterotrophic respiration” and why the authors consider this difference only in different depths of collars digging. References?

Response 3: Thank you for your comment. Line 80-84, in the introduction, the definitions of soil respiration and heterotrophic respiration were added. Total soil respiration refers to the total amount of carbon dioxide released by the respiration of all microorganisms, plant roots, and soil animals within the soil. This includes both heterotrophic and autotrophic respiration. Soil heterotrophic respiration refers to the CO2 produced by the decomposition and metabolism of organic matter by microorganisms and soil animals. In the study area, the predominant vegetation type is alpine meadow, with 90% of plant roots distributed in the surface soil (0-30 cm). The deep ring can exclude 90% of newly grown roots in this portion, thereby eliminating root-related respiration. This method employs the ring method.

References:[41]Subke J A, Voke N R, Leronni V, et al. Dynamics and pathways of autotrophic and heterotrophic soil CO2 efflux revealed by forest girdling[J]. Journal of Ecology, 2011, 99(1): 186-193.

Comments 4: Line 206: “a represents the respiration rate at temperature 0℃”. How this value measured? If it is not measured, from where did the authors take it?

Response 4: Thank you for your comment. Line 236-237, in section 2.5, Q10 is calculated using the van’t Hoff equation (i.e., an exponential function).

References: [44] J.H. van’t Hoff Lectures on Theoretical and Physical Chemistry. Part 1: Chemical Dynamics Edward Arnold, London (1898), pp. 224-229.

Comments 5: Lines 210-211: “b is the respiratory temperature sensitivity coefficient derived from formula (1)." Please describe the formula.

Response 5: Thank you for your comment. Line 244-247, based on the measured soil respiration rate and soil temperature, the exponential equation is derived to obtain parameters a and b. Q10 represents the factor by which the soil respiration rate changes with a temperature increase of 10 ℃. By substituting b into formula (2), Q10 can be calculated.

Comments 6: In Figure 1, the word“measurement”seems missing in“(C) Soil respiration rate in the thermal subsidence area. (D) 132 Soil respiration rate under warming treatment”.

Response 6: Thank you for your comment. Line 139-140, in Figure 1, the word "measurement" has been added. It is revised to: “(C) Measuring soil respiration rate in the thermokarst subsidence area; (D) Measuring soil respiration rate under warming treatment.”

Comments 7: In Figure 4, the error bars visually do not correspond to the letters that indicate significant differences between treatments.

Response 7: Thank you for your comment. There was an error in the data used for plotting Figure 4, and the image has been corrected.

Comments 8: Throughout the text, the control variants of the experiments had their own dynamics, which the authors did not take into account when describing the results. In this regard, I believe that the methodology of the article requires a more complete description and the results a complete revision.

Response 8: Thank you for your comment. I will address your question from the following three aspects.

(1)Experimental Design and Control Group Description: We selected three sites in the Qinghai Lake River Source Wetland, encompassing both natural conditions and thermokarst subsidence areas. Four treatments were established: Control (CK), Warming (W), Thermokarst Subsidence (RR), and a combined treatment of Thermokarst Subsidence and Warming (RRW), with three replicates for each treatment. The control group (CK) was not subjected to warming treatment and was not affected by thermokarst subsidence, serving as a baseline reference to ensure that changes in the other treatment groups could be accurately assessed. During the data monitoring process, soil respiration, soil temperature, humidity, and related environmental factors for all treatment groups were dynamically compared with the control group, aiming to eliminate any interference from natural variations in environmental factors during the experiment.

(2)Use of Open-Top Chambers (OTC) for Warming Simulation: Open-top chambers (OTC) were employed to simulate warming, ensuring the reproducibility and reliability of the experiment. The specific parameters of the OTC devices are detailed, including their warming principles and extensive applications in alpine wetland ecosystems. The control of air turbulence and wind speed within the OTC allows for stable warming effects to be achieved.

Ou Y Q et al. conducted a study published in 2019 in "Grassland and Lawn" titled "Effects of Short-term Simulated Temperature Enhancement on Biomass and Soil Respiration Rate of Subalpine Meadow," where they measured biomass and soil respiration rate in subalpine meadows using OTC warming techniques.

Wang J et al. published a study in 2022 in "Acta Agrestia Sinica" titled "Effects of Seasonal Asymmetric Simulated Warming on Community Characteristics of Alpine Meadows on the Qinghai-Tibet Plateau," where they used OTC warming to explore the impact of warming on plant community species composition and diversity in alpine meadows.

Fu G et al. published a study in 2015 in "Ecology and Environmental Sciences" titled "Response of Community Aboveground Parts Carbon and Nitrogen Content to Experimental Warming in an Alpine Meadow at Three Elevations in Northern Tibet," where they conducted OTC warming experiments in alpine meadows at different elevations to investigate the response of carbon and nitrogen content in the community's aboveground parts to simulated warming.

(3)In the results section, the reviewer noted that the dynamics of soil respiration and related environmental factors in the control group (CK) should be adequately considered. In the revised manuscript, we have further supplemented and emphasized the dynamic comparative analysis between the control group and treatment groups, clearly delineating the independent effects of warming and thermokarst subsidence on soil respiration.

Round 2

Reviewer 2 Report

Comments and Suggestions for Authors

In the manuscript, the methodology of differentiation between “total soil respiration” and “soil heterotrophic respiration" is referenced to the paper Subke J A, Voke N R, Leronni V, et al. Dynamics and pathways of autotrophic and heterotrophic soil CO2 efflux revealed by forest girdling. Journal of Ecology, 2011, 99(1): 186-193. In this paper, Subke et al. describe forest girdling in a mature Western Hemlock (Tsuga heterophylla) and use the girdling to prevent nutrients flowing from tree roots into the soil. The study described in the manuscript was done in the alpine meadow, so the authors should provide more arguments for using this method.

Author Response

Comments 1: In the manuscript, the methodology of differentiation between “total soil respiration” and “soil heterotrophic respiration" is referenced to the paper Subke J A, Voke N R, Leronni V, et al. Dynamics and pathways of autotrophic and heterotrophic soil CO2 efflux revealed by forest girdling. Journal of Ecology, 2011, 99(1): 186-193. In this paper, Subke et al. describe forest girdling in a mature Western Hemlock (Tsuga heterophylla) and use the girdling to prevent nutrients flowing from tree roots into the soil. The study described in the manuscript was done in the alpine meadow, so the authors should provide more arguments for using this method.

Response 1: Thank you for your insightful comment. To address this point, we have carefully reviewed relevant literature and identified similar methodologies that have been successfully applied in alpine meadow ecosystems, despite differences in the study environments.

In a study by Wang et al. (2024) published in Nature Geoscience, titled "Enhanced response of soil respiration to experimental warming upon thermokarst formation," they employed trenching techniques to separate components of soil respiration in thermokarst-affected areas of the Tibetan Plateau, which have marsh meadow vegetation. They used 5 cm and 42 cm PVC rings to distinguish between total soil respiration and heterotrophic respiration, referring to this method as the trenching method. This demonstrates the applicability of trench-based separation techniques in environments beyond forest ecosystems.

Additionally, in 2010, Yan et al. published in Global Change Biology ("Differential responses of auto- and heterotrophic soil respiration to water and nitrogen addition in a semiarid temperate steppe"), where they applied a similar methodology in alpine meadows on the Tibetan Plateau. Shallow PVC rings were used to capture total soil respiration (root and microbial), while deeper rings, from which all plants were removed, isolated heterotrophic respiration by excluding root carbon inputs.

Similarly, Song et al. (2017) in Biogeosciences ("Initial shifts in nitrogen impact on ecosystem carbon fluxes in an alpine meadow: patterns and causes") conducted experiments in alpine meadows using a comparable method. They installed 5 cm rings for total soil respiration and 40 cm rings to measure heterotrophic respiration. The larger rings severed old roots and prevented new root growth, with surface vegetation removed to eliminate root-derived carbon. These rings were installed a year in advance to ensure decomposition of any residual roots, thereby ensuring accurate separation of microbial respiration.

These examples demonstrate that trenching or ring-based methods have been validated in similar high-altitude ecosystems and are an appropriate approach for distinguishing between autotrophic and heterotrophic respiration in alpine meadows. This provides a strong rationale for applying similar methods in our study.

I have revised the content of section "2.4. Soil respiration measurement" based on your suggestions and updated the relevant references. The revisions have been marked in red in the manuscript.